# Turning Garlic into a Modern Crop: State of the Art and Perspectives

**DOI:** 10.3390/plants12061212

**Published:** 2023-03-07

**Authors:** Ricardo Parreño, Eva Rodríguez-Alcocer, César Martínez-Guardiola, Lucía Carrasco, Purificación Castillo, Vicent Arbona, Sara Jover-Gil, Héctor Candela

**Affiliations:** 1Instituto de Bioingeniería, Universidad Miguel Hernández, Campus de Elche, 03202 Elche, Spain; 2Departamento I+D, Coopaman S.C.L., Carretera Peñas De San Pedro, km 1.6, 02006 Albacete, Spain; 3Departament de Ciències Agràries i del Medi Natural, Universitat Jaume I, 12071 Castelló de la Plana, Spain

**Keywords:** garlic, breeding, *Allium sativum*, genetics, genomics, fertility restoration

## Abstract

Garlic is cultivated worldwide for the value of its bulbs, but its cultivation is challenged by the infertility of commercial cultivars and the accumulation of pathogens over time, which occurs as a consequence of vegetative (clonal) propagation. In this review, we summarize the state of the art of garlic genetics and genomics, highlighting recent developments that will lead to its development as a modern crop, including the restoration of sexual reproduction in some garlic strains. The set of tools available to the breeder currently includes a chromosome-scale assembly of the garlic genome and multiple transcriptome assemblies that are furthering our understanding of the molecular processes underlying important traits like the infertility, the induction of flowering and bulbing, the organoleptic properties and resistance to various pathogens.

## 1. General features and challenges

Garlic (*Allium sativum* L.) is a monocotyledonous plant belonging to the family *Amaryllidaceae*, in the order Asparagales. It is native to Central Asia and is cultivated in temperate climates worldwide, with an annual production of 28 million tons on approximately 1.6 million hectares (http://fao.org/faostat/, accessed on 31 January 2023). China and India are the largest producers of garlic, accounting for 80% of the global production (Table 1). With references to its use dating back to ancient Egypt and India 5000 years ago, garlic is one of the oldest known crops.

The genus *Allium* comprises 1018 known species (https://powo.science.kew.org/, accessed on 30 January 2023), including perennial geophytes characterized by the production of bulbs or rhizomes. Bulbs are an adaptation to growth in arid regions with dry summer periods. The genus includes species of great economic and agronomic interest, cultivated for their bulbs and pseudostems, used as food or condiment, such as onion (*A. cepa* L.), shallot (*A. ascalonicum* L.), chives (*A. schoenoprasum* L.), leek (*A. ampeloprasum* L.), bunching onions (*A. fistulosum* L.) and Chinese chives (*A. tuberosum* Spreng.), or for ornamental purposes, such as *A. aflatunense* B. Fedtsch.

Several attempts have been made to reconstruct the phylogeny of the genus [1]. The phylogenies obtained through parsimony and neighbor-joining methods using internal transcribed spacer (ITS) sequences of nuclear ribosomal RNAs were generally congruent with each other [1], with the exception of the placement of certain species, and with previous molecular phylogenies [2]. According to these phylogenies, the development of bulbs in *Allium* species is the ancestral state for this trait, being common in the species of the *Amerallium* and *Allium* subgenera. The presence of rhizomes is considered to be a derived character, while the production of bulbs in onion is considered a reversion to the ancestral state.

In the cultivated species of *Allium*, the basal part of the foliage leaves differentiates to form bulbs and pseudostems, which constitute the edible portion of the leaves. Garlic bulbs are rich in organosulfur compounds, which impart them their unique organoleptic properties [3]. The most abundant of these compounds is allicin (allyl disulfide oxide), which is synthesized from the non-protein amino acid alliin by the enzyme alliinase (EC 4.4.1.4) in response to tissue disruption [4]. Allicin has been used as a food additive due to its multiple bioactive properties [4].

### 1.1. The Life Cycle of a Garlic Plant: Bulb and Flower Development

Some aspects of the anatomy and development of sterile and fertile garlic accessions have been studied in great detail by different authors [5,6,7,8]. Garlic bulbs contain a variable number of cloves, usually ranging from 8 to 14, and are covered by outer tunics (i.e., the dry sheaths of older foliage leaves). The cloves are lateral shoots that develop from the axillary meristems located on the adaxial side of foliage leaves. Each clove is itself a small bulb and is covered by dry, protective leaves, which vary in color (white, purple or reddish) and cover a modified storage leaf that constitutes the fleshy part of the clove. The apical meristem of the clove is located at the apex of a very short stem called the basal plate, and is flanked by the developing leaf primordia. The life cycle of a garlic plant (Figure 1) starts when a clove exits dormancy and sprouts after being exposed to temperatures between 5 °C and 10 °C for several weeks [9,10]. One of the earliest events in the development of a new plant is the emergence of adventitious roots at the periphery of the basal plate, preceding the development of flat foliage leaves. Foliage leaves are initiated from the apical meristem and eventually form a pseudostem. The root system and the leaves usually develop before bulb formation.

Of particular interest are the mechanisms that control the floral transition and the induction of new bulbs. The ability to bolt, whereby the shoot apical meristem shifts from forming vegetative leaves to produce a floral scape, only occurs in some varieties. This trait has been used as a criterion for distinguishing garlic subspecies. Garlic varieties have been classified as having a flowering stalk (“bolting”, “stalking” or “hard neck”), without a flowering stalk (“non-bolting”) or with a partial stalk (“soft neck”) [10]. In the soft-neck cultivars, the plant bolts but the elongation of the stalk is incomplete, and no mature flowers are formed. The inflorescence of garlic is an umbel containing up to 100 flower clusters, or cymes, each with 5 or 6 flowers [8]. The inflorescences of many varieties might also contain topsets (bulbils). The topsets are propagules that, similar to the cloves, allow for the vegetative propagation of the plant and are thought to compete with the flowers for nutrients and space, limiting and sometimes preventing flower development. The flowers are smaller than those of onions, with floral organs arranged in five whorls, including a perianth formed by six tepals (the outer whorls one and two), six stamens (whorls three and four), and a syncarpous ovary formed by three fused carpels that define three locules (whorl five), each containing two ovules [7,11,12,13].

Bulbing and bolting are key developmental transitions that are regulated by temperature and photoperiod. Two stages can be distinguished in relation to the development of the bulbs and the inflorescence [14]. During the so-called inductive stage, short-day (9 to 10 h) photoperiods and low (10 °C) temperatures trigger the flowering transition in many garlic clones [15], as well as the initiation of lateral buds from the leaf axils [13]. Later in the crop season, during the morphogenic stage, the long-day photoperiod and higher temperatures of the spring promote the filling of the cloves, bolting and flower development. The enhancement of bulbing and cloving by low temperatures has been known for a long time [16], and is the basis of the vernalization or “conditioning” treatments that are applied to the cloves before sowing [14]. Conditioning has been reported to accelerate bulbing, allowing an earlier harvest of the bulbs and better bulb formation [14,17]. On the other hand, long-day photoperiods might result in smaller bulbs as a result of accelerated bulbing, while they promote the elongation of the flower stalk [6,18]. The mutual effects of bulbing on bolting and vice-versa are not fully understood. Bulb development is known to initiate earlier in non-bolting cultivars, possibly implying the existence of a balance or competition between bulb and inflorescence development [19], but some studies have failed to appreciate this competition, as stem elongation seemed to have no impact on bulb weight [18]. The optimal temperatures and photoperiods might vary among accessions, as also occurs in other plants [18]. Indeed, flowering in garlic is strongly influenced by the genetic background, as some accessions fail to bolt even under long photoperiod and inductive conditions, which is one of the reasons that contributes to explain the infertility of garlic cultivars.

### 1.2. Consequences of Vegetative Propagation and Fertility Restoration

Garlic is typically propagated vegetatively using the cloves or bulbils from the previous harvest, as the cultivated varieties have lost the ability to reproduce by true seeds (sexual reproduction). The infertility of garlic impedes the making of controlled crosses and limits the implementation of modern breeding methods, which is one of the main challenges that hinders the development of garlic as a modern crop. As a result, new garlic varieties have typically been obtained through clonal selection of spontaneously occurring variants [20]. The accumulation of pathogens over time, which reduces plant vigor, is another major burden associated to the vegetative propagation of garlic. Viral infections can reduce crop yields by up to 50% and are perpetuated through vegetative propagation [21], forcing growers to periodically clean the planting material by applying time-consuming and costly methods to select plants that are free of viruses and other pathogens in an attempt to mitigate this problem [22]. These methods typically involve the micropropagation of shoot apical meristems, although cultures derived from almost every plant organ have been tested, including leaves, basal plates and roots. In vitro cultures are often combined with additional treatments, such as thermotherapy, cryotherapy and chemotherapy, which selectively affect the survival of the infected cells [23,24]. The cost of these procedures is further increased by the need to go from the in vitro cultures to full production capacity, a process that might require several years [19]. Because the bulbils do not develop in direct contact with the soil, they have been proposed to contain a lower pathogen load and are also used in the propagation of garlic [25]. However, a recent study has found that some viruses also accumulate in the bulbils, pointing to the need for additional treatments before bulbils can be effectively used for garlic propagation. The need to treat and store a large amount of cloves until the next growing season is very costly in personnel and resources, which is one of the main factors that decrease the economic yield of garlic [26].

Research on flowering and seed production will lead to major changes in the future of garlic as a crop. The selection of fertile varieties (capable of reproducing by true seed) is expected to solve problems such as the accumulation of pathogens in bulbs and bulbils and would accelerate the breeding of new cultivars with improved characteristics. The production of true seeds in some garlic accessions was first described in the 1950s [27], but the first studies of sexual reproduction and the breeding of fertile varieties were not carried out until the 1980s [12]. Since then, there have been major advances in this field, such as the identification of fertile varieties [28,29,30,31,32]), as well as detailed descriptions of their flowering and reproductive characteristics [6,13,33].

## 2. The Genetics and Genomics of Garlic: State-of-the-Art

Despite the agricultural importance of garlic, the repertoire of resources for garlic breeding used to be very limited. The lack of molecular tools, combined with the asexual mode of reproduction, has made it difficult to use modern marker-assisted selection methods and incorporate the existing genetic variation into breeding programs, which have almost exclusively relied on clonal selection strategies. This situation has left breeders with few options, such as clonal selection, chemical or physical mutagenesis, exploiting somaclonal variation, mutation induction, transgenesis, polyploidy induction and genome editing [34,35,36].

### 2.1. Molecular Markers and Linkage Maps

Prior to genome sequencing, numerous researcher groups reported the development and use of various types of molecular markers, from isozymes [37,38] to DNA-based markers, including RAPD (random amplified polymorphic DNA) [38], SSR (simple sequence repeats, also known as microsatellites) [39,40,41,42,43,44,45,46,47,48,49], ISSR (inter simple sequence repeats) [44,50], SRAP (sequence-related amplified polymorphism) [51] and AFLP (amplified fragment length polymorphisms) markers. These markers have been utilized for various purposes, such as studying the phylogenetic relationships between *Allium* species, characterizing the intraspecific diversity in garlic and related species and assessing the relationships among the accessions of germplasm collections [39]. Some authors have questioned whether the variants of some SSRs actually correspond to true alleles [52]. The alleles of some molecular markers followed Mendelian segregation ratios in plant families obtained through sexual reproduction [52]. With the availability of the garlic genome sequence, the genomic location of these markers can now be determined, which has confirmed the non-allelic character of many randomly selected SSR sequences. Therefore, markers based on expressed sequences, such as ESTs (expressed sequence tags) or assembled RNA-seq reads, are expected to be more reproducible and informative [46,47,48,49,53].

The available toolkit includes several low-resolution linkage maps, which depict the position and relative distance of a limited number of molecular markers [54,55]. The first linkage map of the garlic genome [54] was constructed in 2005 using 84 plants from an S_1_ mapping population, generated by self-fertilization of a single plant of a Soviet-origin variety. The map incorporated 83 single-nucleotide polymorphisms (SNPs) and 8 insertion-deletion markers (indels) developed from expressed sequences, which were assigned to 9 linkage groups. The mapping population also enabled the study of male sterility, which was inherited as a recessive trait and could be assigned to one of the linkage groups. In addition, continuous variation was reported for several traits, suggesting that their genetic architecture can be investigated using conventional quantitative trait locus (QTL) mapping techniques. A second map was reported the same year [55] based on AFLP markers and gene-specific markers, using two mapping populations obtained through self-pollination. The maps obtained from these populations consisted of 20 and 13 linkage groups, respectively, which exceeds the number of chromosomes in the garlic haploid genome (*n* = 8). Both studies ruled out seed production due to apomixis, although this phenomenon had been previously described in some *Allium* species. Instead, the observed Mendelian segregation ratios clearly indicated amphimixis. Furthermore, some markers showed a 15:1 segregation ratio, as expected for duplicated markers, uncovering the existence of duplicated sequences in the garlic genome. Unfortunately, these linkage maps were created using mapping populations that, to our knowledge, have not been maintained by the corresponding research groups, which limits their usefulness because new molecular markers cannot be added and integrated maps incorporating different types of molecular markers cannot be created.

Genotyping a large number of molecular markers in all individuals of a population can be achieved though techniques, such as genotyping by sequencing (GBS) [56] or restriction-site-associated DNA (RAD) sequencing (RAD-Seq) [57], and is a crucial preliminary step for mapping and identifying genes and QTLs relevant for crop breeding. Despite the significant challenge posed by the large size of the garlic genome, GBS methods have been successfully applied to construct at least three linkage maps in closely related species, such as onion [58,59,60]. The first of these maps was made using 96 F_2_ plants from a cross involving 2 different onion cultivars, which contained more than 10,000 single-nucleotide polymorphisms (SNPs) spanning the 8 chromosomes of the onion haploid genome [58]. The main challenge in constructing linkage maps in garlic is the infertility of most available varieties, which prevents the making of crosses and mapping populations. However, DArT-seq (Diversity Array Technology sequencing), a method related to GBS, has been successfully used in garlic to evaluate the genetic diversity and to identify redundant accessions in a large germplasm collection from Spain [61]. In this work, the authors identified 131 redundant accessions (out of 417 accessions), which allowed them to select a core set that captures the genetic diversity of the collection. However, Barboza and colleagues [48] have recently warned against the elimination of accessions on only the grounds of marker genotypes, emphasizing the importance of retaining epigenetic variants, and furthermore, pointing out that the germplasm collection studied by Giménez and García-Lampasona had higher levels of epigenetic variation than of genetic variation [62,63]. The successful use of DArT-seq implies that similar GBS methods could also be used to create high-resolution linkage maps in garlic. In many species, these maps were made using recombinant inbred line (RIL) or doubled haploid (DH) mapping populations, which represent permanent resources for gene mapping and offer unlimited material for building integrated linkage maps incorporating different types of markers [64,65]. An additional advantage of these populations is that they can be characterized multiple times, under different environmental conditions or at different geographical locations for QTL mapping. The ability to vegetatively propagate garlic offers an easy and affordable alternative to the generation of RIL or DH populations, as the progeny of any cross can be perpetuated with no need to perform further crosses.

### 2.2. Transcriptomic Approaches in Garlic

The garlic genome is diploid (2*n* = 2x = 16), with an estimated size 5 times larger than the human genome, reaching 15.9 gigabases [66]. This large size is a common property among other *Allium* species, making it challenging to study. Prior to genome sequencing, some studies provided information on the transcriptome (the expressed part of the genome) of various tissues, such as meristems and other organs [67,68,69,70,71]. In addition to efforts to characterize the gene content of the garlic genome, efforts have also been made since 2015 to identify genes in phylogenetically close species, such as onion [72], bunching onions [73] and Chinese chives [74].

The study of garlic genes will further our understanding, at the molecular level, of the main problems faced by this crop, such as the infertility. Some transcriptomic studies aimed at understanding its molecular basis and might help to restore the production of true seeds in some varieties [8,71]. Other studies have focused on the relationship between storage temperatures and bulb formation [75]. The development of new molecular markers [76,77,78,79] and their application to the genetic improvement of these new fertile varieties promises to be a revolution in the development of this crop. The first garlic transcriptome assemblies were performed in 2012 using Illumina sequencing technology [68]. The de novo assembly of reads from two libraries, prepared from dormant and germinating vegetative buds from field-collected bulbs, allowed the assembly of 127,933 unigenes, which were assigned functions by comparing the sequences with different databases and assigning gene ontology (GO) terms with the BLAST2GO program. The obtained sequences were used to identify genes involved in the sulfur assimilation pathway as well as in the biosynthesis of organic sulfur compounds. In 2015, Kamenetsky and co-workers [70] assembled and annotated an integrated transcriptome of six organs (flowers, inflorescences, leaves, cloves, basal plate and root) of a fertile garlic variety, which allowed them to identify orthologues of genes involved in flowering and sulfur metabolism, as well as to detect the presence of some viruses that were present in the sequenced samples. A comprehensive summary of efforts to characterize the transcriptome in garlic is presented in Table 2.

### 2.3. The Genomes of Garlic and Related Species

We are currently witnessing a revolution in the number of sequenced genomes for *Allium* species, with available sequences for the genomes of garlic [85], onion [87] and bunching onion [88]. Garlic was the first *Allium* species to have a chromosome-scale genome assembly, which was achieved through a combination of PacBio, Illumina and ONT sequencing technologies, as well as 10X Genomic libraries and Hi-C technology. The assembly yielded a 16.24 Gb sequence, spanning 96.1% of the genome according to estimates based on *k*-mer statistics. The genome contained 57,561 annotated protein-coding genes, 10% of which are located in tandem repeats, and 20,008 non-coding RNA genes, including 3741 miRNAs, 8984 tRNAs, 4352 rRNAs and 2931 snRNAs. The relatively low BUSCO (Benchmarking Universal Single-Copy Orthologs) values suggested that this assembly still requires further improvement, which might be achieved using optical and chromosome-contact maps developed using Bionano Genomics technology.

The main difficulty in the assembly of the garlic genome lies in its enormous complexity, given that it contains a high number of repetitive sequences (14.8 Gb, approximately 91.3% of the genome assembly, higher than the percentage in the maize genome), clustered in the central regions of the chromosomes, and high levels of heterozygosity. The large number of transposable elements (which account for 76% of the genome size) contribute to its large size and their insertions often affect the expression of other genes. The large size of the garlic genome has been attributed to the expansion of *gypsy*-type LTR retrotransposons [88]. In fact, 4219 garlic genes were found to be disrupted by transposon insertions, including an orthologue of the floral homeotic gene *APETALA2*.

At the time of its release in 2020, the garlic genome was the largest one for a monocotyledonous plant, although six additional genomes in the same order (*Asparagales*) were also available [85]. The phylogenetic analysis of these genomes showed that garlic forms a monophyletic clade with *Narcissus viridiflorus*, a member of the *Amaryllidaceae* family. Garlic was phylogenetically closest to *Asparagus officinalis*; the two species shared a common ancestor 80.8 million years ago (mya). In plants, the expansion of gene families occurs by polyploidization caused by whole-genome duplication (WGD) events and the proliferation of transposable elements. Analysis of the synteny between garlic and *A. officinalis* uncovered 3 WGD events in the evolutionary history of garlic, which occurred 120–130, 89.8 (prior to the divergence of garlic and *Asparagus*) and 17.9 mya. Two transcriptome assemblies have recently been performed using third-generation sequencing technologies [83,86], which promise to lead to a better annotation of the garlic genome. 

The genome of bunching onion consists of 8 chromosomes and is 11.27 Gb in size, which roughly matches the size estimated from *k*-mer statistics (11.97 Gb). This genome contains 62,255 genes and has high levels of repeated sequences (89.89% according to *k*-mer statistics, and 69.81% according to repeatmasker). The collinearity between the genomes of *A. fistulosum* and *A. cepa* is higher than that between the genomes of *A. fistulosum* and garlic. In the latter, the existence of genomic rearrangements is apparent, although the number of chromosomes in the haploid genome remains at eight. This correlation was already apparent in linkage maps with SSR-type markers. The onion genome is diploid (2*n* = 16), consisting of as many chromosomes, as the genome of *A. fistulosum*. It has been determined that the repetitive fraction of the onion genome amounts to 95%, being especially rich in *copia* and *gypsy* retrotransposons [87]. Genome sequencing was performed with a doubled haploid accession combining PacBio and Illumina reads. The availability of linkage maps allowed anchoring scaffolds to as many pseudo-molecules as there are chromosomes in the genome. However, not all scaffolds could be oriented in the linkage map. The proportion of repetitive sequences turned out to be lower than initially expected. Annotation will require additional effort, as the article documents the prediction of 540,925 gene models, of which only a much smaller fraction has experimental support in RNA-seq reads.

## 3. Target Traits for Garlic Breeding

### 3.1. Yield and Bulb Traits

Considerable variation among clones has been reported, including differences in flower stalk formation, bulbil development in inflorescences, bulb size and morphology and bulb color. Other traits of agronomic interest include number of cloves, organoleptic properties (flavor, pungency and color), health-enhancing effects (nutraceutical value), clove uniformity, firmness, early harvest, late bolting, resistance to diseases, and differences in the production of phenolic and sulfur compounds [18,48,89,90,91,92]. Breeding strategies in garlic have been primarily oriented toward the improvement in bulb yield and quality. Continuous variation has been reported for traits such as bulb size (height, width and weight), number of cloves and number of days from sowing to harvest [49]. Not unexpectedly, a significant correlation was found between bulb size components and number of cloves.

### 3.2. Secondary Metabolism as a Target for Garlic Breeding

The health-promoting and flavor attributes of garlic have been associated with the presence of different secondary metabolites, including polyphenolic compounds, organosulfur metabolites (alkenyl cysteine sulfoxides, S-allyl cysteine, thiosulfinates, diallyl sulfides, vinyldithiins and different isomers of ajoene) [3] and oligosaccharides, primarily acting as storage carbohydrates in bulbs [93,94]. Volatile sulfur-containing compounds (namely allicin, which accounts for ~70% of the total thiosulfinate compounds produced after crushing garlic cloves) are important contributors to flavor traits of garlic and are produced by the decomposition of S-alkenyl cysteine sulfoxides (alliin, methiin, propiin and isoalliin) following an enzyme-catalyzed process. Moreover, allicin is highly unstable and can be spontaneously converted into different allyl sulfides that contribute to the flavor of garlic products [3]. At the biochemical level, isoalliin and S-carboxypropyl-cysteine sulphoxide are derived from glutathione, whereas alliin derives from the reaction of the amino acid serine with an allyl thiol of unknown origin and a glutamate moiety rendering g-glutamyl-S-allyl-L-cysteine, which is then metabolized by a flavine-containing monooxygenase a sulfoxide and subsequently by a GTPase rendering the corresponding alliin [95].

Secondary metabolites influence organoleptic properties such as color, pungency and taste, and might also constitute the end-products of the domestication process to adapt to different environmental conditions, hence acting as actual markers of geographic origin of *Allium* varieties [93]. Fructo-oligosaccharides are important storage compounds present in onion bulbs, and organosulfur compounds are key determinants of the aroma of *Allium* species. Flavonoids are primarily involved in the color of the bulb (particularly in onions) and also play an important role in abiotic and biotic stress tolerance [93]. In this regard, the inner peels (enclosing each clove) and outer peels (enclosing the entire bulb) of garlic bulbs do have different metabolic composition: inner clove peels are rich in organic acids such as gluconic, gulonic acids and vanillic acid, whereas outer bulb peels are richer in carbohydrates (rhamnose, lyxose, glucose, xylobiose D and trehalose) along with sugar alcohols (mannitol, sorbitol and threitol), both differing greatly from clove metabolite composition. Some of these metabolites likely account for the allelopathic properties of onion bulb peels [96]. Liu and co-workers [97] analyzed the evolution of metabolites in different parts of the garlic plant (leaf, pseudostem, bulb peel wrapper, clover skin and clove) and identified 84 different compounds whose dynamics were highly correlated with the storage role of bulbs.

The production of non-pungent, tear-free onions is a successful example of new variety development by manipulating the secondary metabolism of organosulfur compounds in an *Allium* species. The garlic genome contains 60 genes of the alliinase gene family [85], which undoubtedly contribute to its organoleptic properties. In onion, the tear-inducing lachrymatory factor is synthesized through two consecutive, enzymatically catalyzed reactions. The first reaction is catalyzed by an alliinase enzyme activity, while the second is catalyzed by the lachrymatory factor synthase (LFS). Although onions are recalcitrant to transformation, the LFS gene was silenced using a transgenic RNAi approach—the first example of gene silencing in onions [98]. More recently, tearless onions have also been selected in a mutant screen after heavy-ion beam mutagenesis. In this case, bulbs with reduced alliinase activity were identified in M_3_ families derived from 1,450 M_1_ irradiated seeds [99].

## 4. Pathogens of Garlic: Threats, Treatments and Perspectives

An undesirable consequence of vegetative propagation is the accumulation of pathogens in garlic bulbs, which causes a reduction in yield and an increase in costs. Many different types of pathogens affect garlic, including arthropods (such as insects and mites), fungi, bacteria, viruses and phytoplasmas. Various procedures can be used for their control, ranging from the sanitation of planting material using sophisticated in vitro culture procedures, to crop rotation, avoiding the use of the same plots in consecutive years and the use of phytosanitary products to control the dispersal of vectors. Current restrictions on the use of these substances pose difficulties in maintaining product quality, so it is essential to identify germplasm accessions that are genetically resistant, or at least less susceptible, to attack by different types of pathogens and to open the door to the use of the new biotechnological tools available.

### 4.1. Fungal Pathogens

Fusarium basal rot (FBR), the most devastating soil-borne disease of garlic, is caused by the necrotoph fungus *Fusarium oxysporum* f. sp. *cepae* (FOC) [100]. The first visible symptoms of FBR are the curling and yellowing of the leaves, eventually leading to rotting of the basal part or the whole plant. Importantly, the damage caused by FBR eventually becomes the entry point for other secondary pathogens. Promising approaches to prevent the proliferation of fungal pathogens involve the use of fungicides, intercropping, soil solarization techniques and storing the bulbs under optimal conditions of humidity and temperature, but the use of resistant accessions remains the most cost-effective approach to control fungi and other soil-borne pathogens [91,101,102]. The AsRGA29 (*A. sativum* resistant gene analog 29) was found to be induced after inoculation with FOC in a resistant garlic line (named CBT-As153), or in other *Allium* species that have been reported to be naturally resistant to the fungus, such as *A. fistulosum* and *A. roylei*, as well as after exogenous treatments with methyl jasmonate, salicylic acid, abscisic acid and hydrogen peroxide, suggesting its putative involvement in the response against FBR [91]. Sequencing and quantitative PCR have uncovered families of microRNAs (miRNAs) involved in the activation of the immune response of garlic against FOC [103]. The miR394 microRNA was also found to be responsive to FOC inoculation, but the induction was more pronounced in a sensitive line (CBT-As11) than in the resistant line (CBT-As153) [104]. In line with this observations, two predicted targets of miR394, which encode an F-box protein and a P450 cytochrome, were expressed at lower levels after FOC inoculation, and both the expression of miR394 and its targets were regulated by methyl jasmonate [104]. Another disease, Fusarium dry rot (FDR), is caused by *Fusarium proliferatum* and mainly affects the crop during the post-harvest bulb storage [105]. *F. proliferatum* and *F. oxysporum* do not have a specific host [106], and intercropping with species that can be infected but are naturally resistant to these pathogens, such as maize and spring wheat [107], decreases inoculum levels in the soil [108]. Other gene families putatively involved in defense mechanisms against *Fusarium* infections have been identified in garlic, including genes that encode TLP (thaumatin-like proteins) [109], PR (pathogenesis-related) proteins [110] or CHI (class I chitinase) proteins [101], whose expression is positively correlated with the response to *Fusarium* in plants resistant to the fungus.

*Alternaria porri* (the causal agent of purple blotch) [111], *Sclerotium cepivorum* (white rot) [112], *Stemphylium* spp. (leaf blight) [113], *Botrytis allii* (neck rot) [114], *Aspergillus* spp. (black mold), *Penicillium corymbiferum* (decay) [115] and *Puccinia spp.* (garlic rust) [116] are fungi that cause serious diseases at distinct steps of cultivation. Most of the research done to identify sources of resistance to these fungal infections has been performed in species other than garlic, but the information gained will be highly valuable once fertility has been restored. The resistance of onion to purple blotch was studied by crossing resistant and sensitive cultivars, and was inherited as a monogenic, dominant trait. Bulked segregant analysis suggested that two markers were linked to the locus conferring pathogen resistance [117]. *Agrobacterium*-mediated transformation of calli has also been performed to express tobacco proteins (chitinase and glucanase) against *S. cepivorum*, leading to a reduction of invasion in transformed garlic plants [118]. To identify sources of resistance to *Stemphylium*, allotetraploid hybrids were created between *A. cepa* and *A. fistulosum*, as well as recombinant chromosomes between both species following consecutive crossings [119]. Time-series expression analysis unveiled differential expression of several genes related to the jasmonic acid signaling pathway (*JAR1*, *COI1*, and *MYC2*) after *Botrytis* infection in two onion lines (one resistant and one susceptible), suggesting that these genes participate in the plant response against *Botrytis* [120]. In regard to *Puccinia*, a recent study using monosomal alien addition lines (MAAL), which add chromosomes from rust-resistant *A. cepa* varieties to *A. fistulosum* plant, suggested that resistance genes reside on chromosome 1 of onion [121].

### 4.2. Nematodes

Nematodes such as *Ditylenchus dipsaci* cause wilting, chlorosis and damaged or rotted bulbs, as they facilitate the entry of other pathogens, such as *Fusarium* strains that cause FDR or FBR [122,123]. The nematode reproduction rate (FR) has been evaluated in different garlic varieties, finding that some varieties have low FR both in vitro and in the field [124]. Although one cultivar from Israel was completely resistant, its bulbs were not suitable for commercialization [125]. Recently, it has been proposed that a specific PTI (Pattern-triggered immunity) response to nematode attack involves DNA hypomethylation in the CHH context in certain regions of the genome, altering the expression patterns of some plant genes [126].

### 4.3. Arthropods: Insects and Mites

Arthropods represent a problem because they damage the plants and the bulbs at the growing and post-harvest stages and because they act as vectors for viral pathogens. The eriophyid mite *Aceria tulipae* (dry bulb mite) invades plants causing a characteristic curling of the leaves. At the storage stage, the mites absorb sap from the clove, forming brown and reddish spots that greatly reduce bulb weight [127]. In addition, the mites transmit different viruses of genus *Allexivirus* [128,129,130]. Bolting varieties of garlic have been reported to be more susceptible to mite attack than non-bolting ones [131]. Insects from several orders, such as *Ephestia cautella* (Lepidoptera), *Delia antiqua* (Diptera) or *Thrips tabaci* (Thysanoptera), feed on the plants or stored bulbs and cause important yield losses [132,133]. Moreover, garlic breeding would significantly benefit from the identification of resistance sources to these pests.

### 4.4. Viruses

Plant pathogenic viruses reduce yield by affecting bulb weight and diameter, leading to significant economic losses worldwide. Plants are often simultaneously infected by multiple viruses of the *Allexivirus*, *Potyvirus* and *Carlavirus* genera (all three characterized by having single-stranded positive-sense RNA genomes [134]), synergistically causing yield losses [132]. Genus *Allexivirus* includes *Garlic virus A* to *Garlic virus E* and *Garlic virus X* (GarV-A to GarV-E and GarV-X), *Shallot Virus X* (ShVX) and *Garlic Mite-borne Filamentous Virus* (GarMbFV) [135]. Allexiviruses are transmitted by *Aceria tulipae* [129], and the effects of their infection might range from asymptomatic to leaf mosaics and reduced bulb size. Genus *Potyvirus* include viruses such as *Onion yellow dwarf virus* (OYDV), *Leek yellow stripe virus* (LYSV), *Shallot Yellow Stripe Virus* (SYSV) and *Tobacco etch virus* (TEV) [136,137,138,139]. Genus *Carlavirus* includes the *Garlic yellow mosaic-associated virus* (GYMaV) [140], the *Garlic common latent virus* (GarCLV) and the *Shallot latent virus* (SLV; a synonym for *Garlic latent virus*, GLV), the latter two generating chlorotic ringspots with a necrotic center distributed on the leaf [141]. All these potyviruses and carlaviruses are transmitted by aphids.

Other viruses that infect *Allium* species belong to the genera Nepovirus (Artichoke yellow ringspot virus, AYRV), Cucumovirus (Cucumber mosaic virus, CMV), Necrovirus (Leek white stripe virus, LWSV), Tospovirus (Irish yellow spot virus, IYSV) and Fijivirus (Garlic dwarf virus, GDV) [139,142,143]. The use of new massively parallel sequencing technologies facilitates the identification of viruses in plants showing symptoms of virosis. The Illumina technology, for example, has been used to detect Garlic virus E [144].

### 4.5. Bacteria and Phytoplasmas

In addition to viruses, some bacteria also cause diseases in garlic during development, at harvest time or upon storage. During garlic development, *Xanthomonas axonopodis* pv. *allii* causes whitish, lenticular-shaped, brown lesions that spread on the leaves. In episodes of advanced infection, it causes progressive leaf death and reduces bulb size [145]. Another disease that occurs during development is garlic blight, caused by phytoplasmas, which alters the color of the leaves. The color change begins at the leaf apices and moves toward the base, ultimately causing the death of the plant [146]. At harvest time, the most common disease is bacterial soft rot caused by *Erwinia carotovora* subsp. *carotovora*, *Dickeya chrysanthemi*, *Pectobacterium carotovorum* subsp. *carotovorum* and *Lactobacillus* spp. In the early stages of infection, symptoms appear in the neck region of garlic and cause a watery rot in the infected tissues [147]. Upon storage, infections with *Pseudomonas salomonii* and *Pseudomonas fluorescens* cause a unique phenotype consisting of dark brown spots on the tunics of garlic bulbs, known as *café au lait* [148,149]. So far, garlic immunity mechanisms towards viral and bacterial infections have not been described.

## 5. Opportunities for the Future of Garlic Breeding

Knowledge of the genome and genes involved in the development of an organism, such as garlic, or in the development of a certain phenotypic trait of interest, represent a fundamental step toward its detailed functional characterization at the molecular level. Understanding the function of garlic genes is expected to facilitate the cultivation under adverse environmental conditions, such as high temperature, increased soil salinity or drought, without affecting its development or the crop’s yield. Understanding the natural variation would make it possible to determine the contribution of certain alleles to beneficial phenotypic traits. Sexual reproduction will enable the introduction of such traits into elite cultivars through controlled crosses. The clonal propagation of garlic is associated with a low level of intra-population genetic diversity, making the populations more prone to diseases, which make garlic less likely to successfully cope with the harsh conditions of a changing climate or with emerging pathogens.

In the last decades, many authors have made impressive efforts to generate the tools required for the development of improved garlic cultivars. Major advances included the establishments of large collections of molecular markers, and the demonstration that linkage maps of garlic can be built with the available tools. The most important milestone is the restoration of seed production in certain garlic lines that had retained the ability to reproduce sexually. Extensive knowledge on the environmental conditions that trigger bulb and inflorescence development has accumulated in recent year. The development of massively parallel sequencing technologies in the last decade has been instrumental to gain rapid access to the genetic information of plants with enormous genomes, such as garlic. These technologies initially allowed the de novo assembly of garlic transcriptomes, but the advent of long-read sequencing technologies is now facilitating the assembly of complex genomes, such as those of Allium crops, which are rich in various types of repeats. There are countless existing protocols that have addressed different aspects of garlic tissue culture, transformation mediated by *Agrobacterium tumefaciens* or by biolistic methods and plant regeneration. Altogether, these methods and the available genome sequences will open the door to the use of novel genome editing tools in garlic. The possibility of obtaining segregating populations of garlic by sexual reproduction, combined with the ability to maintain these populations indefinitely through vegetative reproduction, will greatly facilitate the study of the genetic architecture of quantitative traits of agronomic interest, such as those described in this review.

## Figures and Tables

**Figure 1 plants-12-01212-f001:**
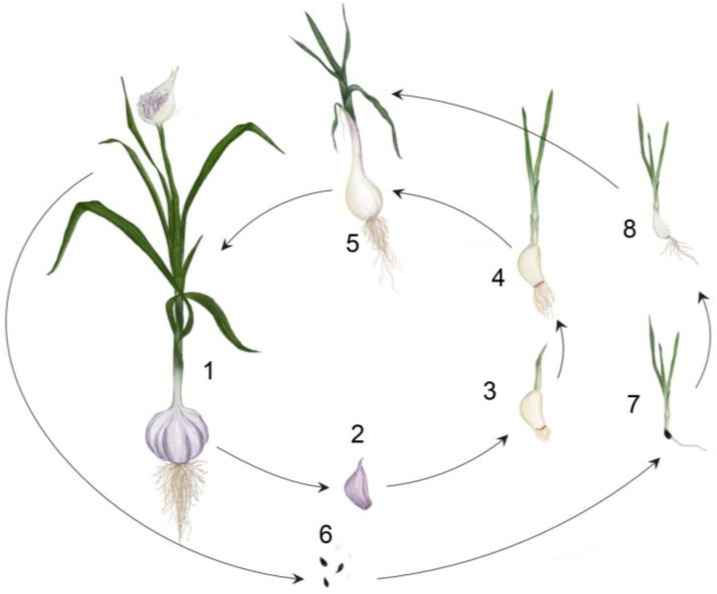
Life cycle of a garlic plant showing the different stages from the germination of a seed or clove to a mature plant. (1) Adult garlic plant of a bolting cultivar bearing a floral scape. (2) Dormant clove, recently detached from the mother plant, which has not yet produced adventitious roots. (3) Sprouted clove in which new adventitious roots and the first leaves have emerged. (4) Young garlic plant in which the formation of new cloves has not yet been initiated. (5) Garlic plant in which the bulb is actively thickening as a result of the growth of new cloves. (6) True seeds are produced only in some fertile garlic varieties. (7) When a seed germinates, a new seedling emerges, with a primary root emerging from the embryo’s apical meristem and new leaves developing from the shoot apical meristem. (8) Under inductive conditions, seed-derived plants can also produce new bulbs, which can then be vegetatively propagated if desired.

**Table 1 plants-12-01212-t001:** Global production of *Allium* crops in 2021.

Crop	Cultivated ha ^1^	Tons Produced	Yield (Tons/ha)	Highest Producers
Garlic	1,659,236	28,204,854	17.00	China, India, Bangladesh, Egypt
Leeks and other alliaceous vegetables	134,168	2,213,183	16.49	Indonesia, Turkey, Belgium, France
Onions and shallots, dry	5,778,767	106,592,008	18.44	India, China, Egypt, United States
Onions and shallots, green	215,933	4,665,525	21.60	China, Mali, Japan, Republic of Korea

^1^ Data according to FAOSTAT (http://fao.org/faostat/, accessed on 31 January 2023).

**Table 2 plants-12-01212-t002:** Transcriptome studies using garlic.

Reference	Sequencer	Software	Sample	Results
Kim et al., 2009 [67]	Sanger		Leaf and stem tissues	21,595 ESTs
Sun et al., 2012 [68]	Illumina HiSeq 2000	SOAPdenovo	Dormant and sprouting vegetative buds	127,933 unigenes
Sun et al., 2013 [69]	Illumina HiSeq 2000		See [68]	45,363 DEGs *
Kamenetsky et al., 2015 [70]	Illumina MiSeq	Trinity	Inflorescence, flower, leaf, clove, roots and basal plate	239,116 (‘extensive’), or 102,042 contigs (‘abundant’ transcriptome)
Shemesh-Mayer et al., 2015 [71]	Illumina Hiseq 2000	Bowtie, DESeq	Flower buds at 3 developmental stages	16,271 DEGs *
Liu et al., 2015 [46]	Illumina HiSeq 2500	Trinity, MISA	10 days old plants, 45 days old	135,360 unigenes; 1506 SSR markers
Havey and Ahn, 2016 [52]	Sanger and Roche 454-FLX	SOAPdenovo-trans	Leaf, pseudostem and root tissues	35,936 contigs; 14,879 SNP and indel markers
Zhu et al., 2017 [80]	Illumina HiSeq 2500	Trinity	Leaf tissue	132,225 unigenes
Chaturvedi et al., 2018 [81]	Illumina HiSeq 2000	Bowtie, DESeq	Internal buds and storage leaves at two temperatures	8303 (internal buds) and 14,147 DEGs * (storage leaves)
Li et al., 2018 [82]	Illumina HiSeq 4000	Trinity, CD-HIT	Cloves stored at 4°C for 0, 10, 15 and 40 days	49,280 unigenes; 5923 DEGs
Chen et al., 2018 [83]	PacBio RSII (Iso-Seq CCS) and Illumina HiSeq 2500	proovread, CD-HIT	Developing bulb	36,321 transcripts
Liu et al., 2020 [84]	Illumina HiSeq 2500	Trinity, edgeR	Stem (control and treated with GA3)	159 DEGs *
Sun et al., 2020 [85]	Illumina HiSeq 2500	Trinity, DEGseq	Sprouts, bulbs, flowers, roots, pseudostems and leaves	34,439 transcripts with constitutive (28,394) or specific (964) expression (out of 57,561 genes predicted in the genome)
Wang et al., 2022 [86]	PacBio Sequel (CCS)	Quiver, CD-HIT-EST	Lower bulb, aerial bulb, scape, leaf, clove, basal plate and roots	36,571 high-quality consensus reads

* Differentially Expressed Genes.

## Data Availability

No new data were created or analyzed in this study. Data sharing is not applicable to this article.

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
