# Peer review of "Turning Garlic into a Modern Crop: State of the Art and Perspectives"

_plants, 2023, doi:10.3390/plants12061212_

Round 1

Reviewer 1 Report

Dear Authors,

The submitted manuscript titled „Turning garlic into a modern crop: state-of-the-art and perspectives” contains interesting results. Nevertheless, I have found some imperfections, which in my opinion should be imroved or clarified before an eventual publication. Please, find them below.

The major issues

1.      In my opinion the  manuscript should be divided into main parts such as Introduction, Material and methods, Results, Discussion, Conclusions.

2.       In part Introduction the bacground, purpose and main aims of ivestigations should be outlined. The ecological characteristics of genus (morphology, life span, range, habitat affiliation) should be provided in Introduction or in chapter Studied genus.

3.      In manuscript there is no description of methods of literature search. The procedure of literature search should be based on PRISMA statements. They are detaildly presented in publications of Moher et al. such as:

·         Page M J, McKenzie J E, Bossuyt P M, Boutron I, Hoffmann T C, Mulrow C D et al. The PRISMA 2020 statement: an updated guideline for reporting systematic reviews BMJ 2021; 372 :n71 doi:10.1136/bmj.n71

·         David Moher, Alessandro Liberati, Jennifer Tetzlaff, Douglas G. Altman, 2010. Preferred reporting items for systematic reviews and meta-analyses: The PRISMA statement, International Journal of Surgery, 8(5), 336-341.

4.      I encourage Authors to add at least short Discussion the results with other literature sources.

Author Response

We would like to thank the reviewer for his/her feedback and the time to review our manuscript. Although we understand the reviewer's concern about the lack of PRISMA criteria in our review, we would like to emphasize that our focus was not to conduct a systematic review, but rather to provide an overview of the existing literature. Despite this, our review was comprehensive and rigorous, and we believe it provides valuable insights into the topic.

For this reason, we have chosen not  to follow the reviewer's advice, as we believe our approach was appropriate given the scope of our work. We note that systematic reviews are more appropriate in a different context (e.g. metaanalysis of clinical trials), and that the NCBI's Pubmed classification of manuscripts clearly distinguishes between "reviews" and "systematic reviews".

Reviewer 2 Report

The manuscript is interesting and well written. Authors highlighted recent developments in garlic, including genome and transcriptomic studies, problems in reproduction and resistance to pests, diseases and viruses. Minor review is needed. Comments are included into pdf version.

Author Response

We would like to thank the reviewer for his/her positive assessment of our manuscript. We have made an effort to address all the points raised by the reviewer, which he/she added as comments to the PDF file. We believe that these changes have help us to improve our manuscript. All the changes are highlighted in the revised version of the manuscript, which we have resubmitted.

We have also taken this opportunity to cite the tables and figures at appropriate places in the text, and have corrected some errors in Table 2, which we have also made more concise.

Best regards,

Hector Candela

Round 2

Reviewer 1 Report

Dear Authors,

In my opinion the manuscript is sufficiently corrected and might be publish in present form.